# Enhanced Sensitivity of A549 Cells to Doxorubicin with WS_2_ and WSe_2_ Nanosheets via the Induction of Autophagy

**DOI:** 10.3390/ijms25021164

**Published:** 2024-01-18

**Authors:** Weitao Jin, Ting Yang, Jimei Jia, Jianbo Jia, Xiaofei Zhou

**Affiliations:** 1College of Science & Technology, Hebei Agricultural University, Huanghua 061100, China; jin1631112@163.com (W.J.);; 2Institute of Environmental Research at Greater Bay Area, Key Laboratory for Water Quality and Conservation of the Pearl River Delta, Ministry of Education, Guangzhou University, Guangzhou 510006, China; 3Hebei Key Laboratory of Analysis and Control of Zoonotic Pathogenic Microorganism, Baoding 071000, China

**Keywords:** WS_2_, WSe_2_, autophagy, chemosensitizer, cancer therapy

## Abstract

The excellent physicochemical properties of two-dimensional transition-metal dichalcogenides (2D TMDCs) such as WS_2_ and WSe_2_ provide potential benefits for biomedical applications, such as drug delivery, photothermal therapy, and bioimaging. WS_2_ and WSe_2_ have recently been used as chemosensitizers; however, the detailed molecular basis underlying WS_2_- and WSe_2_-induced sensitization remains elusive. Our recent findings showed that 2D TMDCs with different thicknesses and different element compositions induced autophagy in normal human bronchial epithelial cells and mouse alveolar macrophages at sublethal concentrations. Here, we explored the mechanism by which WS_2_ and WSe_2_ act as sensitizers to increase lung cancer cell susceptibility to chemotherapeutic agents. The results showed that WS_2_ and WSe_2_ enhanced autophagy flux in A549 lung cancer cells at sublethal concentrations without causing significant cell death. Through the autophagy-specific RT^2^ Profiler PCR Array, we identified the genes significantly affected by WS_2_ and WSe_2_ treatment. Furthermore, the key genes that play central roles in regulating autophagy were identified by constructing a molecular interaction network. A mechanism investigation uncovered that WS_2_ and WSe_2_ activated autophagy-related signaling pathways by interacting with different cell surface proteins or cytoplasmic proteins. By utilizing this mechanism, the efficacy of the chemotherapeutic agent doxorubicin was enhanced by WS_2_ and WSe_2_ pre-treatment in A549 lung cancer cells. This study revealed a feature of WS_2_ and WSe_2_ in cancer therapy, in which they eliminate the resistance of A549 lung cancer cells against doxorubicin, at least partially, by inducing autophagy.

## 1. Introduction

Worldwide, lung cancer is one of the most common cancers and the main cause of cancer-related deaths. There are an estimated 2 million new cases and 1.76 million deaths of lung cancer annually [1]. Non-small-cell lung cancer (NSCLC) accounts for approximately 85% of newly diagnosed lung cancer patients [2]. Despite substantial progress in the treatment of NSCLC, the survival rates of these patients have not been significantly improved in the past few decades [3]. Therefore, new therapeutic approaches urgently need to be established. Recently, therapeutic strategies based on nanoparticles (NPs) have shown significant advantages in cancer treatment, surpassing conventional drug delivery [4]. There are many types of NPs, among which two-dimensional transition-metal dichalcogenides (2D TMDCs) are a typical 2D NP. Their high specific surface area and photothermal conversion coefficient endows them with good application prospects in the field of biomedicine. Existing studies have shown that 2D TMDCs possess desirable features in drug delivery, photothermal and electrothermal therapy, photo-dynamic therapy, biological imaging, and biosensing [5]. Moreover, 2D TMDCs have been developed for the ablation of cancer cells [6] and the modulation of the tumor microenvironment [7]. However, these studies were still in their infancy, and the detailed molecular mechanisms need to be explored.

As a process conserved from yeast to humans, autophagy participates in cell survival and death by sequestering, transporting, and degrading cytoplasmic components [8]. Recent studies have found that autophagy is closely related to various cell death types, such as apoptosis and ferroptosis. The fact that the tumor suppressor protein p53 [9], BH3-only proteins [10], Akt, JNK, and DAPK [11] regulate both autophagy and apoptosis may explain the sequential activation of these two processes. Autophagy stimulates apoptosis through the activation of key apoptotic proteins. For example, the inhibition of autophagy by 3-methyladenine (3-MA) inhibited the activation of caspase 3 and caspase 9, which indicated the key role of autophagy in the activation of caspases [12]. In addition, ferroptosis is a newly discovered form of regulatory cell death, which depends on the accumulation of intracellular iron and subsequent lipid peroxidation [13]. Autophagy may contribute to iron overload or lipid peroxidation, and eventually cause ferroptosis. For example, ferroptosis induced by Fe_4_O_3_ NPs is possibly regulated through Beclin1/ATG5-dependent autophagy [14]. Autophagy itself is closely related to the survival and death decision of cells. As an external stimulus, NPs can elevate cell autophagy to extremely high levels, and in most cases, promote cell death [15]. For example, ZnO NPs increased the LC3-II/LC3-I ratio in human osteosarcoma cells. Pre-treatment with 3-MA to inhibit autophagosome formation alleviated ZnO NP-induced cell death, verifying the pro-death role of autophagy [16]. Other NPs, such as quantum dots [17] and up-conversion NPs [18], induced pro-death autophagy.

Chemosensitizers can improve the cytotoxicity of anticancer chemotherapeutic agents, although their own anticancer activity is very low [19]. The commonly used chemosensitizers include agents that reverse multidrug resistance, deliver siRNA molecules, and target hypoxia, and those used as Chinese herbal medicine [20]. Simultaneously, new chemosensitizers are being developed to promote effective cancer therapy, especially for malignant and multidrug-resistant tumors [21,22]. Recent research indicated that nanoscale nitric oxide delivery effectively reprogrammed the immune microenvironment and tumor vasculature, overcoming resistance to cancer therapy [23]. This emphasizes the ability of NPs to enhance the sensitivity of cancer cells to chemotherapeutic agents. Two-dimensional TMDC-based cancer theranostics, such as chemosensitization, have been explored [24]. However, the mechanisms of 2D TMDC-mediated chemosensitization have not yet been thoroughly studied. We previously showed that the cell surface adhesion and cellular internalization of MoS_2_ and WS_2_ induced mTOR-dependent autophagy through interactions with IGF-1R, APP, or CXCR4 in 16HBE cells [25,26]. We recently demonstrated that 2D TMDC-induced outer chalcogen-dependent autophagy caused MH-S cells to be more susceptible to respirable benzo[a]pyrene. Autophagy is closely related to cell death. Here, we aimed to investigate whether WS_2_- and WSe_2_-induced autophagy could sensitize non-small-cell lung cancer cells (A549) to chemotherapeutic drugs. Our comprehensive data indicated that WS_2_ and WSe_2_ pre-treatment with sublethal doses significantly increased the sensitivity of A549 cells to the chemotherapeutic agent doxorubicin (DOX) via autophagy induction. Our research was conducted in vitro, and the results are a preliminary exploration of the chemotherapy sensitization effect of 2D TMDCs to facilitate the development of 2D TMDC-based cancer nanotherapeutics.

## 2. Results and Discussion

### 2.1. Characterization of WS_2_ and WSe_2_ Nanosheets

The cellular functional disturbance caused by NPs is influenced by their physicochemical properties. For 2D NPs, thickness and size are important physicochemical characteristics. Previous studies have found that the size of graphene oxide nanosheets affected the polarization of macrophages and the production of inflammatory cytokines [27]. The thickness of MoS_2_ nanosheets caused genetic disturbance in 16HBE cells [25]. The morphology and thickness of WS_2_ and WSe_2_ were characterized via atomic force microscopy (Figure 1a–f). The WS_2_ and WSe_2_ exhibited a similar thickness of 5.0 nm and comparable lateral sizes that ranged from 300 to 500 nm. Both the WS_2_ and WSe_2_ nanosheets carried negative charges, as shown by their zeta potentials (Appendix A). In water, the zeta potentials of WS_2_ and WSe_2_ are −39.0 ± 1.9 mV and −41.2 ± 1.5 mV. However, in the cell culture medium containing 10% FBS, the zeta potentials of WS_2_ and WSe_2_ were shifted to −11.3 ± 1.2 and −11.7 ± 0.9 mV (Appendix A), indicating protein adsorption on the nanosheet surface.

### 2.2. Cellular Localization of WS_2_ and WSe_2_


The subcellular localization of NPs directly affects their perturbation of intracellular homeostasis. Previously, we found that in 16HBE pulmonary bronchial epithelial cells, thinner MoS_2_ and WS_2_ nanosheets more frequently bound to the cell membranes and were less likely to enter the cell interior [25,26]. In mouse alveolar macrophage MH-S cells with strong phagocytic activity, thin WS_2_, WSe_2_, NbS_2_, and NbSe_2_ nanosheets not only attached to the cell membrane, but were also observed inside cells, such as in lysosomes and the cytoplasm [28]. To explore the interaction between WS_2_ and WSe_2_ nanosheets and A549 cells, transmission electron microscopy (TEM) was performed to analyze the subcellular localization of WS_2_ and WSe_2_ after incubation with A549 cells at an exposure dose of 25 μg/mL for 24 h. As shown in Figure 1, the WS_2_ and WSe_2_ nanosheets were more frequently observed to adhere to the cell membranes of the A549 cells (Figure 2a,c), with a small number entering the cell interior (Figure 2b,d). This result is consistent with previous studies indicating that thinner 2D nanosheets, such as graphene nanosheets [29], polymeric nanosheets [30], and Mg(OH)_2_ nanosheets [31], have a strong tendency to bind to the cell surface. Our results also found an intracellular localization of a small number of nanosheets, which may be due to the aggregation of thinner 2D nanosheets into 3D nanoaggregates, making them prone to be internalized by cells [25]. The cell surface adhesion and intracellular localization of WS_2_ and WSe_2_ increased the possibility that they would interact with cell surface receptors and intracellular proteins to perturb cell signaling pathways and cell functions.

### 2.3. Autophagy Induction by WS_2_ and WSe_2_

Both WS_2_ and WSe_2_ exhibited very low cytotoxicity at a concentration of 50 μg/mL in the A549 cells (Appendix A). A low dose of WS_2_ and WSe_2_ might increase cellular stress and induce cell autophagy. To access the autophagy perturbation by WS_2_ and WSe_2_ nanosheets, TEM was used to observe the autophagosome formation. The TEM examinations showed that both WS_2_ and WSe_2_ exposure resulted in the formation of autophagosomes (Figure 3a–c), indicating that both nanosheets induced autophagy in A549 cells. In addition to the qualitative observation of autophagosomes, the LC3-II protein levels in the A549 cells exposed to WS_2_ or WSe_2_ were determined quantitatively. The protein levels of LC3-II in the A549 cells exposed to 5 μg/mL of WS_2_ or WSe_2_ were 2.4 and 1.3 times that of the basal LC3-II levels in cells treated with the cell culture medium, respectively, showing the enhanced autophagy in the A549 cells (Figure 4a,b,e,f). The quantitative analysis showed dose-dependent LC3-II formation after the WS_2_ and WSe_2_ treatments.

The accumulation of autophagosomes and the increased expression of LC3-II protein may be a result of an enhanced formation of autophagosomes or an impaired degradation of autophagosomes. Existing studies have shown that NPs weaken the lysosome degradation capacity or inhibit the fusion between autophagosomes and lysosomes by causing lysosome alkalization, lysosomal overload, and cytoskeleton disruption, thereby blocking autophagy flux [32,33,34]. To clarify the main cause of autophagy perturbation in response to WS_2_ or WSe_2_ exposure, an autophagy flux assay was performed by measuring the expression of p62, which is selectively degraded in autolysosomes and serves as an indicator for autophagic degradation [35]. The results showed that the formation of p62 induced in cells treated with WS_2_ or WSe_2_ at 25 μg/mL was 28% and 16% lower, respectively, than in cells treated with the culture medium (Figure 4c,d,g,h). These data demonstrate that WS_2_ and WSe_2_ enhanced the autophagy flux rather than blocking the degradation of autophagosomes in A549 cells. A similar induction of autophagy was also observed in 16HBE cells and MH-S cells in response to pristine WS_2_ nanosheet exposure, in HeLa (human cervical cancer) and MCF-7 (human breast cancer) exposed to PEGylated WS_2_ nanosheets, and in MH-S cells treated with pristine WSe_2_ nanosheets [26,28,36]. This suggests that the autophagy induction ability of the tested nanosheets was non-selective, and remains to be further confirmed in future studies.

The lithium ions used in 2D nanosheet preparation have induced autophagy in various cell lines [37,38]. The expression of LC3-II induced by the supernatant of a nanosheet suspension was examined to clarify the contribution of lithium ions to WS_2_- and WSe_2_-induced autophagy in A549 cells. Our results showed that no obvious increase in LC3-II protein levels was detected in A549 cells treated with the supernatant of WS_2_ and WSe_2_ suspensions (Appendix A). This suggested that autophagy induction was caused by WS_2_ or WSe_2_ nanosheets, not the supernatant of the nanosheet suspensions.

### 2.4. Autophagy Induction by WS_2_ or WSe_2_ through Perturbation of Different Autophagy-Related Genes

To further reveal the possible mechanisms of WS_2_- and WSe_2_-induced autophagy, we examined the expression of 84 autophagic genes via an autophagy-specific RT^2^ Profiler PCR Array. The mRNA expression levels of 84 autophagic genes in A549 cells after treatment with WS_2_ and WSe_2_ were compared to those in cells treated with the cell culture medium alone. Although WS_2_ and WSe_2_ enhanced autophagy to similar levels, the analysis showed that these two nanosheets activated both common and different autophagy-related genes. Autophagy-related genes such as *APP*, *BAD*, *DRAM1*, *HSP90AA1*, *MAP1LC3A*, and *SNCA* were downregulated, and *HGS* was upregulated by both WS_2_ and WSe_2_ (Figure 5a,b and Appendix A). Furthermore, the expression levels of genes such as *ATG9B*, *DAPK1*, *TNF*, and *ULK1* were upregulated, and *HSPA8* was downregulated by the WS_2_ nanosheets (Figure 5a and Appendix A). After WSe_2_ treatment, the expression level of certain genes, such as *BCL2*, was upregulated, while the levels of *ATG10*, *ATG4A*, *ATG4C*, *ATG4D*, *ATG5*, *CLN3*, *CTSB*, *DRAM2*, *EIF2AK3*, *ESR1*, *GABARAP*, *GABARAPL1*, *GABARAPL2*, *HDAC1*, *MAPK14*, *MAPK8*, *PIK3C3*, *SQSTM1*, *TMEM74*, and *TNFSF10* were downregulated (Figure 5b and Appendix A).

To systematically explore the roles of significantly altered genes in WS_2_- and WSe_2_-induced autophagy, we further constructed a molecular interaction network using the Cytoscape software 3.9.1 based on the KEGG database [39]. Alterations in *APP*, *TNF*, and *mTOR* with WS_2_ and *APP*, *MAPK14*, and *HSP90AA1* with WSe_2_ showed a much higher interdependence among the obviously changed genes (Figure 6a,b), suggesting their central roles in inducing autophagy. NPs may affect cell functions by interacting with cell surface proteins and cytoplasmic proteins. For example, iron oxide NPs and silver NPs promoted autophagy through the activation of the PI3K/AKT/mTOR signaling pathway in HT22 cells [38]. Carbon nanotubes were shown to regulate cell functions by interacting with insulin growth factor-1 receptor [40] and bone morphogenetic protein receptor [41]. WS_2_ and WSe_2_ were observed adsorbed on the cell surface and inside cells, and induced autophagy at a comparable level (Figure 4a–h), which strongly suggests that WS_2_ and WSe_2_ may trigger autophagy by perturbing cell surface receptors and cytoplasmic proteins.

APP, as an integral membrane protein, plays a core role in the pathogenesis of Alzheimer’s disease [42]. The downregulation of amyloid precursor protein (APP) caused by siRNA transfection reduced extracellular regulated protein kinase activation [43], thereby enhancing cell autophagy [44]. Here, we found that both WS_2_ and WSe_2_ downregulated the expression of *APP* (Figure 5a,b and Appendix A), enhancing autophagy. *TNF* was also upregulated by WS_2_ (Figure 5a and Appendix A). Previous reports showed that TNF induced autophagy through RIPK3-dependent AMPK activation in L929 cells [45]. In addition to *APP*, both *MAPK14* and *HSP90AA1* were also altered after WSe_2_ treatment (Figure 6b). Earlier studies showed that the active metabolite of irinotecan and glucose triggered MAPK14-dependent autophagy in HeLa cells [46]. The binding of the cell surface protein HSP90AA1 to avibirnavirus VP2 induced autophagy via the inhibition of the AKT-mTOR pathway in early infection [47]. Similarly, the downregulation of HSP90AA1 by HSP90AA1-siRNA transfection reduced the levels of autophagy in HGF cells [48]. Here, we also found that WSe_2_ upregulated the expression of *MAPK14* and downregulated the expression of *HSP90AA1* (Figure 5b and Appendix A), and induced autophagy (Figure 4b,d,f,h).

The PCR array data also showed that Atg family genes, including *ATG 9B*, *ATG 5*, and *ATG4C*, were also altered after exposure to WS_2_ or WSe_2_. ATG 9B was shown to be involved in vacuole formation during autophagy initiation [49]. The upregulation of ATG 9B contributed to ^125^I radioactive particle-induced autophagy in Huh7 and Hep3B cells [50]. ATG 4C, a cysteine peptidase, is essential for the lipidation and delipidation of LC3. The downregulation of ATG4C through miR-142-3p transfection or the knockdown of *ATG4C* suppressed the autophagy flux in RAW 264.7 cells [51]. WSe_2_ upregulated the expression of *ATG 4C* (Figure 5a and Appendix A), promoting the formation of LC3-II during the autophagy process [52]. Meanwhile, WSe_2_ upregulated the expression of *ATG 5* (Figure 5a and Appendix A), resulting in the expansion of autophagosomes [53]. 

### 2.5. Enhanced Sensitivity of A549 Cells to DOX by WS_2_ and WSe_2_ Nanosheets

In addition to apoptosis, necrosis, and ferroptosis, autophagy-dependent cell death is also an important mode of cell death. Dendrobium officinale polysaccharide-, cinnamaldehyde-, and tea polysaccharide-induced autophagy mediated cell death through the ROS-ATP-AMPK signaling pathway or epigenetic modifications in CT26 cells [54] and gastric cancer cells [55]. Our findings indicated that WS_2_ and WSe_2_ induced autophagy by perturbing autophagy-related genes, implying that the use of these two nanosheets may make A549 cells more sensitive to chemotherapy drugs. To verify this hypothesis, the viabilities of A549 cells with WS_2_ or WSe_2_ nanosheet pre-treatment, followed by the addition of DOX, were carefully compared. DOX was chosen as it is a classical chemotherapy drug for treating human lung cancer [56]. To avoid overwhelming cytotoxicity caused by DOX, concentrations of 25 and 50 μM were used. WS_2_ and WSe_2_ alone did not cause significant cell death (Figure 7a), which is similar to the cytotoxicity assessment shown in Appendix A. The WS_2_- and WSe_2_-treated cells were more sensitive to DOX (25, 50 μM), with a further 6−44% decline in cell viability compared to DOX-treated cells. (Figure 7a). Collectively, our data revealed the WS_2_- and WSe_2_-mediated sensitization of A549 cells to cytotoxicity induced by DOX. To demonstrate that the enhanced sensitivity of cancer cells to DOX mediated by WS_2_ and WSe_2_ can cross cell lines, human glioma cell line U87 cells were further validated. WS_2_ and WSe_2_ alone did not cause significant cell death at a concentration of 50 μg/mL in U87 cells (Appendix A). Compared with U87 cells treated with DOX, cells treated with WS_2_ and WSe_2_ were more sensitive to DOX, and the cell viability further decreased by 10–26% (Appendix A). Nonetheless, further research is needed to improve the efficacy of WS_2_ and WSe_2_ sensitization in killing cancer cells.

### 2.6. WS_2_ and WSe_2_ Sensitized A549 Cells to DOX by Triggering Autophagy

To explore the role of autophagy in the enhanced sensitivity of A549 cells to DOX by WS_2_ or WSe_2_, we first investigated whether the inhibition of autophagy influenced the increased cell death rate with WS_2_ or WSe_2_. The classic autophagy inhibitor 3-MA was used to suppress autophagy [57]. As shown in Figure 7b, there was no significant cell death in the groups exposed to WS_2_, WSe_2_, or 3-MA alone. As expected, we observed that WS_2_ and WSe_2_ treatment significantly increased the number of dead cells when compared with DOX alone. Notably, the inhibition of autophagy by 3-MA partially reversed the enhanced cytotoxicity by 21% and 13% compared to cells without 3-MA treatment in response to DOX with WS_2_ or WSe_2_, respectively. The statistical analysis confirmed significant differences in survival rate data due to 3-MA. We acknowledge that the addition of 3-MA did not have a strong effect on alleviating cytotoxicity. Previous studies have shown that nanosheets can trigger various intracellular biological events, such as oxidative stress [58] and inflammatory responses [59]. We speculated that, in addition to autophagy, various intracellular biological events were involved in the enhanced sensitivity of A549 cells to DOX induced by WS_2_ and WSe_2_ nanosheets. Therefore, the inhibition of autophagy did not show a dominant effect on alleviating cytotoxicity. Nevertheless, these results indicate that autophagy may be involved in the WS_2_- or WSe_2_-mediated sensitization of A549 cells to cytotoxicity induced by DOX. Previous studies have shown a close relationship between autophagy and apoptosis [60]. Autophagy can induce apoptosis [61]. For example, silver NPs induced apoptosis through the enhancement of autophagy via the PI3K/Akt/mTOR signaling pathway in HT22 cells [38]. Zinc oxide NPs promoted cell apoptosis by inducing autophagy in liver cancer cells [62]. To clarify the relationship between apoptosis and the enhanced sensitivity of A549 cells to DOX induced by WS_2_ and WSe_2_, we further investigated the activation of apoptosis signaling at the molecular level. Apoptosis is a programmed cell death process controlled by a caspase signaling cascade. Caspase 3 is the central effector, and the cleavage of caspase 3 indicates the execution of apoptosis. We analyzed the cleavage of caspase 3 through a Western blot. Our data showed that, compared with cells treated with DOX, pre-treatment with WS_2_ or WSe_2_ induced more significant caspase 3 cleavage (Appendix A). As a result, WS_2_ and WSe_2_ may enhance cell apoptosis by inducing autophagy, contributing to the observed sensitivity of A549 cells to DOX in the present work.

## 3. Materials and Methods

### 3.1. Reagents and Antibodies

WS_2_ and WSe_2_ nanosheets were obtained from XFNANO (Nanjing, China). Rapamycin and 3-methyladenine were purchased from Sigma-Aldrich (St. Louis, MO, USA). Primary antibodies against LC3B and p62 were purchased from Cell Signaling Technology (Boston, MA, USA). The primary antibody β-actin was purchased from Abgent (San Diego, CA, USA). Polyvinylidene difluoride (PVDF) membranes and Western blot luminescence reagents were purchased from Millipore (Billerica, MA, USA). 

### 3.2. Cell Culture

The A549 and U87 cell lines were kindly provided by Professor Bing Yan from Guangzhou University. The A549 cells were cultured in an RPMI-1640 cell culture medium (Gibco, Grand Island, NY, USA). The U87 cells were cultured in a DMEM cell culture medium (Gibco, Grand Island, NY, USA). The cell culture medium was supplemented with 10% fetal bovine serum (Gibco, Grand Island, NY, USA) and a 1% penicillin–streptomycin solution (Gibco, Grand Island, NY, USA). The cells were maintained in a humidified cell incubator at 37 °C with 5% CO_2_ for cultivation. They were subcultured every other day and seeded in dishes or plates at an appropriate density for experiments. 

### 3.3. Cellular Localization of WS_2_ and WSe_2_ Determined through Transmission Electron Microscopy

The A549 cells were treated with WS_2_ or WSe_2_ (25 μg/mL) for 24 h. Then, the cells were rinsed and fixed in 3% glutaraldehyde (pH 7.4). Next, the A549 cells were stained with a 2% aqueous uranyl acetate solution and subjected to a concentration gradient of alcohol. After treatment with propylene oxide and propylene oxide/Epon dilutions, the A549 cells were embedded in 100% Epon. The ultrathin sections were cut using an LKB-V ultramicrotome. Images were acquired using a JEOL-1200EX with MORADA-G2 (Agilent, Santa Clara, CA, USA).

### 3.4. Cell Viability Assay

The cell viability of the A549 cells treated with WS_2_ and WSe_2_ was analyzed using a Cell Counting Kit-8 (CCK-8) assay (Yeasen Biotech, Hong Kong). The A549 cells were exposed to WS_2_ or WSe_2_ at a series of concentrations, and the cell viability was measured after 24 h of incubation. In short, the culture medium was removed and each well was rinsed with PBS. The CCK-8 solution diluted with the 1640 culture medium (*v*/*v* = 1:15) was added to the plates. After incubation at 37 °C for 2 h, the absorbance was measured using an ELISA microplate reader (Model 680, Bio-Rad, Hercules, CA, USA) at a wavelength of 450 nm.

The experimental design to evaluate the enhanced sensitivity of A549 cells to DOX with WS_2_ and WSe_2_ nanosheets was as follows: the A549 cells were pre-treated with WS_2_ or WSe_2_ for 24 h. The culture medium was then removed, and the cells were co-exposed to DOX with WS_2_ or WSe_2_ for another 24 h. Specifically, in the cell culture medium, only WS_2_ or WSe_2_ was present in the first 24 h, while both DOX and WS_2_ or WSe_2_ were present in the second 24 h.

The experimental design to evaluate the role of autophagy in the enhanced sensitivity of A549 cells to DOX with WS_2_ and WSe_2_ was as follows: the A549 cells were treated with WS_2_ or WSe_2_ at 50 μg/mL for 24 h. After removing the culture medium, the cells were exposed to DOX (50 μM) with WS_2_ (50 μg/mL) or WSe_2_ (50 μg/mL), with or without 3-MA (100 μM), for 24 h. Then, the viability of the A549 cells was determined.

### 3.5. Western Blot Analysis

The A549 cells were seeded in 60 mm cell culture dishes. After incubation for 24 h, the cells were exposed to WS_2_ or WSe_2_ in the cell culture medium at different concentrations (0, 5, 25, 100 μg/mL) for another 24 h. The treated A549 cells were rinsed with PBS, harvested, and lysed with a cell lysis buffer (FNN0011, Invitrogen, Carlsbad, CA, USA) containing phenylmethylsulfonyl fluoride (P7626, Sigma-Aldrich, St. Louis, MO, USA) and a protease inhibitor cocktail (P8340, Sigma-Aldrich). The lysed samples were centrifuged to obtain protein samples. A BCA Protein Assay kit (Beyotime Biotechnology, Wuhan, China) was used for protein concentration quantification. Proteins with the same concentration were separated via sodium dodecyl sulfate polyacrylamide gel electrophoresis (SDS-PAGE) with different gel percentages (for β-actin, LC3, and p62 with small molecular weights, a 12% separation gel was used. For all the proteins, a 5% stacking gel was used). Then, the separated proteins were transferred onto a PVDF membrane. The membrane was blocked with 5% *w*/*v* defatted dry milk in TBST (TBS with 0.05% Tween-20) for 1 h at room temperature. After washing with TBST, the PVDF membrane was incubated with the primary antibody at 4 °C overnight. Subsequently, the membrane was washed with TBST three times and incubated with the secondary antibody for 1 h at room temperature. Then, the PVDF membrane was washed with TBST another three times and incubated with a luminescent reagent. The protein bands were visualized and the band intensity was quantified using ImageJ.

### 3.6. The Effect of the Supernatant of WS_2_ and WSe_2_ Suspensions on Autophagy

To rule out the possibility of autophagy induced by the supernatant of WS_2_ or WSe_2_ suspensions, the suspensions were separated into WS_2_ or WSe_2_ nanosheets and supernatant through centrifugation (13,000 rpm/min, 20 min). Then, the supernatant was incubated with A549 cells for 24 h at a concentration equivalent to the WS_2_ or WSe_2_. The protein level of the autophagy marker LC3-II in the treated A549 cells was measured using a Western blot assay. 

### 3.7. PCR Array Analysis

After incubating with WS_2_ or WSe_2_ (25 μg/mL) for 24 h, the A549 cells were lysed with Trizol (Invitrogen, Carlsbad, CA, USA), and the total RNA was extracted. About 1.5 μg of RNA was used for the synthesis of cDNA using a SuperScript III Reverse Transcriptase (Invitrogen, CA, USA). cDNA was added to wells of an autophagy-specific PCR array plate (PAHS-084A, Qiagen, Netherlands, Germany) with an autophagy-specific gene or housekeeping gene primer mix pre-loaded in each well. Then, the PCR plate was subjected to a two-step RT^2^ PCR program (95 °C for 10 min, 95 °C for 15 s, and 60 °C for 60 s). The mRNA expression levels of 84 autophagy-specific genes were quantified using the ΔΔCt method.

## 4. Conclusions

In this study, we revealed that WS_2_ and WSe_2_ nanosheets both adhered to the surface of A549 cells and entered the interior of the cells. This increased the opportunity for WS_2_ and WSe_2_ to interact with cell surface proteins and cytoplasmic proteins, thereby inducing cell autophagy. PCR array investigations characterized the expression signature of autophagy-related genes affected by WS_2_ and WSe_2_. By constructing a molecular interaction network, the core regulatory genes of WS_2_- and WSe_2_-induced autophagy were identified, revealing that WS_2_ and WSe_2_ activated autophagy pathways in distinct ways. By leveraging this mechanism, WS_2_ and WSe_2_ pre-treatment significantly enhanced the sensitivity of A549 cells to the chemotherapeutic drug DOX. The addition of the autophagy inhibitor 3-MA partially reversed the enhanced cytotoxicity caused by DOX with WS_2_ or WSe_2_. In summary, this study provides further insight into the molecular basis of the enhanced sensitivity of A549 cells to DOX caused by WS_2_ and WSe_2_ from the perspective of autophagy. Therefore, our study provides support for WS_2_- or WSe_2_-based cancer nanotheranostics, which may also represent an important strategy to reduce the side effects of chemotherapeutic drugs and overcome the drug resistance of cancer cells. Although the induction of autophagy with the tested nanosheets appeared to be non-selective in a variety of cell lines, whether the enhanced sensitivity of A549 cells to DOX of WS_2_ and WSe_2_ nanosheets may also be extended to other NSCLC cells remains to be elucidated. Given that our research was conducted entirely in vitro and the types of cell lines used were limited, further research is needed in vivo to clarify the specific role of autophagy in the WS_2_ and WSe_2_ nanosheet-enhanced sensitization of tumors to chemotherapeutic drugs.

## Figures and Tables

**Figure 1 ijms-25-01164-f001:**
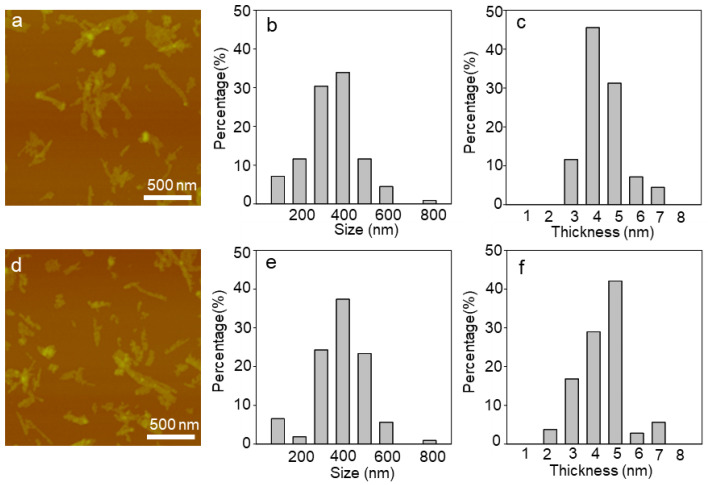
Characterization of WS_2_ and WSe_2_. (**a**,**d**) Typical atomic force microscopy (AFM) images of WS_2_ (**a**) and WSe_2_ (**d**). Distribution of lateral size (**b**,**e**) and thickness (**c**,**f**) of WS_2_ and WSe_2_.

**Figure 2 ijms-25-01164-f002:**
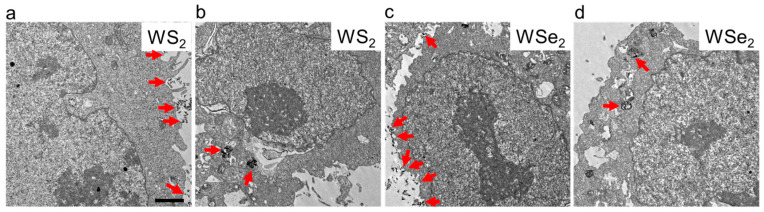
Interactions of WS_2_ and WSe_2_ with A549 cells. After being treated with WS_2_ or WSe_2_ at 25 μg/mL for 24 h, A549 cells were fixed and sectioned for transmission electron microscopy (TEM) observation. The TEM images showed that WS_2_ and WSe_2_ nanosheets adhered to the cell surface (**a**,**c**) or were internalized by cells (**b**,**d**). The narrow red arrow indicates the subcellular localization of WS_2_ or WSe_2_. The scale bar represents 1.5 μm.

**Figure 3 ijms-25-01164-f003:**
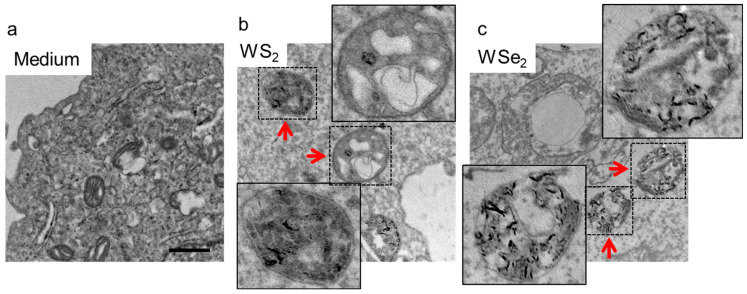
Autophagy induction by WS_2_ and WSe_2_. TEM images of A549 cells post-incubation with cell culture medium (**a**), WS_2_ (**b**) or WSe_2_ (**c**) showed autophagosome formation in response to 2D TMDCs exposures at a concentration of 25 μg/mL for 24 h. The inset images represent the enlargement of single autophagosome. The red arrows indicate the autophagosomes. The scale bar represents 500 nm.

**Figure 4 ijms-25-01164-f004:**
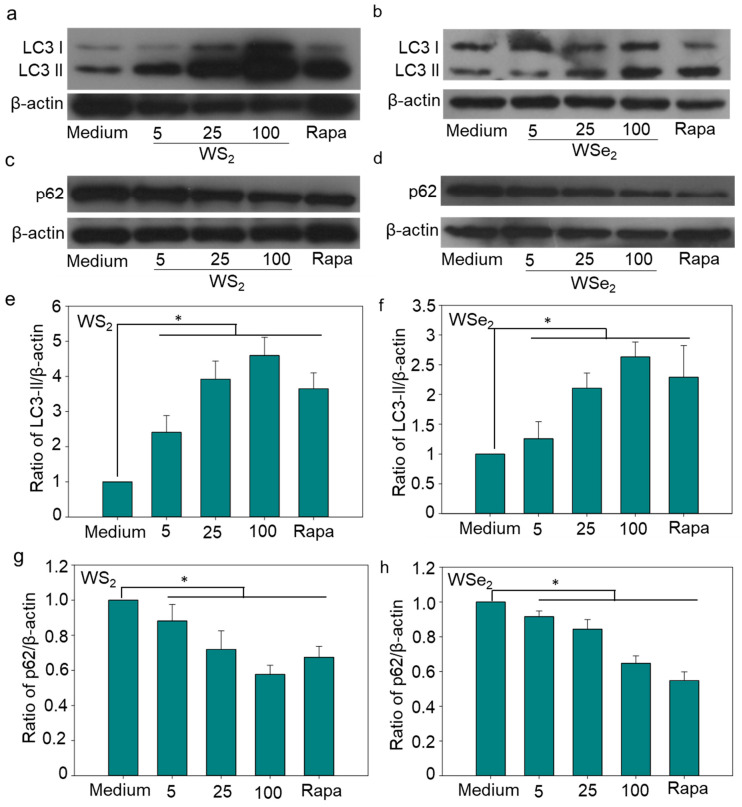
Enhanced autophagy flux induced by WS_2_ and WSe_2_. (**a**,**b**) Dose-dependent LC3-II formation in A549 cells after treatment with WS_2_ (**a**) and WSe_2_ (**b**) as determined via Western blotting against the LC3B antibody. A549 cells treated with cell culture medium or rapamycin (4 μM) were used as negative or positive controls. (**c**,**d**) Expression of p62 protein in A549 cells exposed to WS_2_ and WSe_2_. Cells treated with cell culture medium or rapamycin (4 μM) were used as negative or positive controls. (**e**,**f**) Dose-dependent LC3-II formation in A549 cells quantified based on the intensity ratio of LC3-II to β-actin using ImageJ. Data are shown as mean ± SD (n = 3). (**g**,**h**) Dose-dependent p62 reduction in A549 cells, quantified based on the intensity ratio of p62 to β-actin using ImageJ 1.5.0. Data are shown as mean ± s.d.; n = 3; * *p* < 0.05.

**Figure 5 ijms-25-01164-f005:**
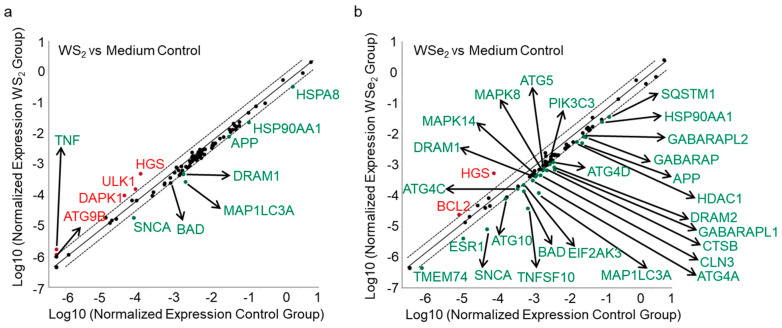
Different perturbations of autophagy-related genes by WS_2_ and WSe_2_. PCR array analysis of the expression of autophagy-related genes in A549 cells after WS_2_ (**a**) or WSe_2_ (**b**) (25 μg/mL) treatment for 24 h compared to that of the cell culture medium treatment.

**Figure 6 ijms-25-01164-f006:**
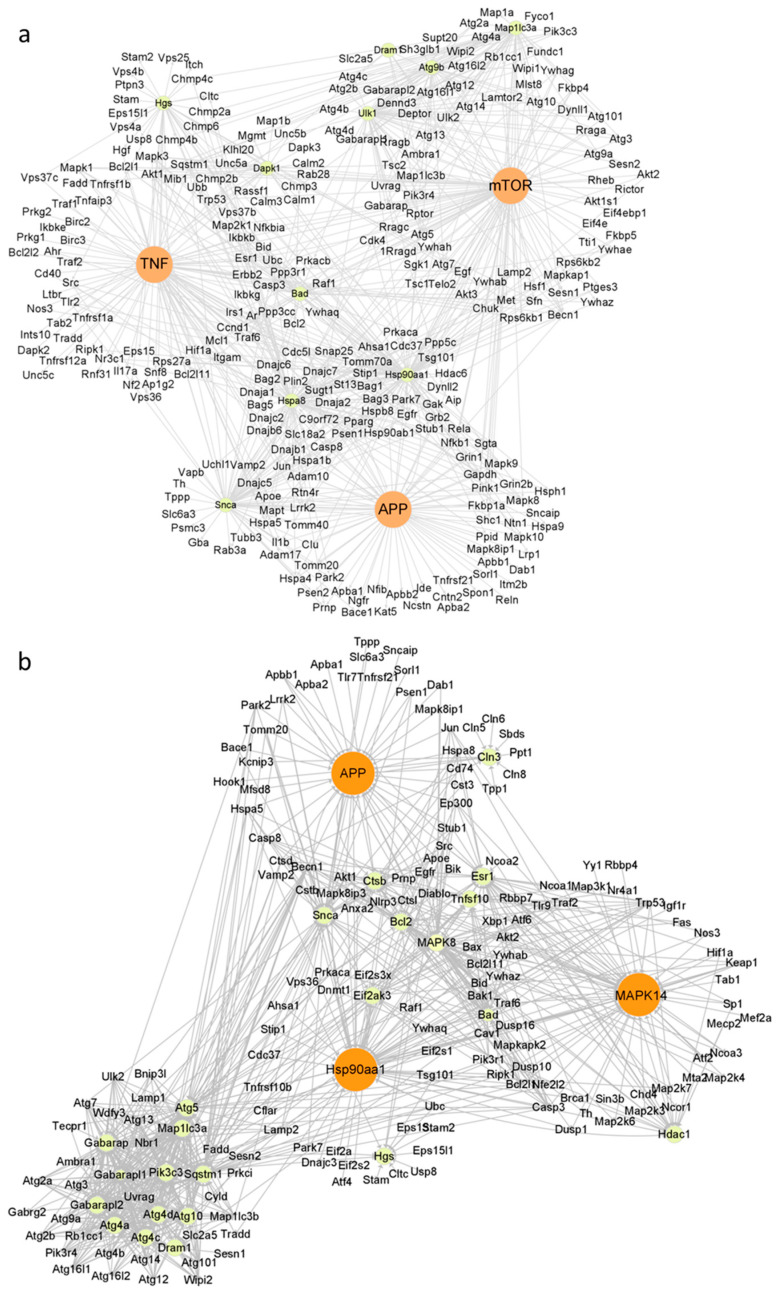
Molecular interaction analysis of autophagy-related genes altered by WS_2_ (**a**) or WSe_2_ (**b**) treatment using Cytoscape software 3.9.1.

**Figure 7 ijms-25-01164-f007:**
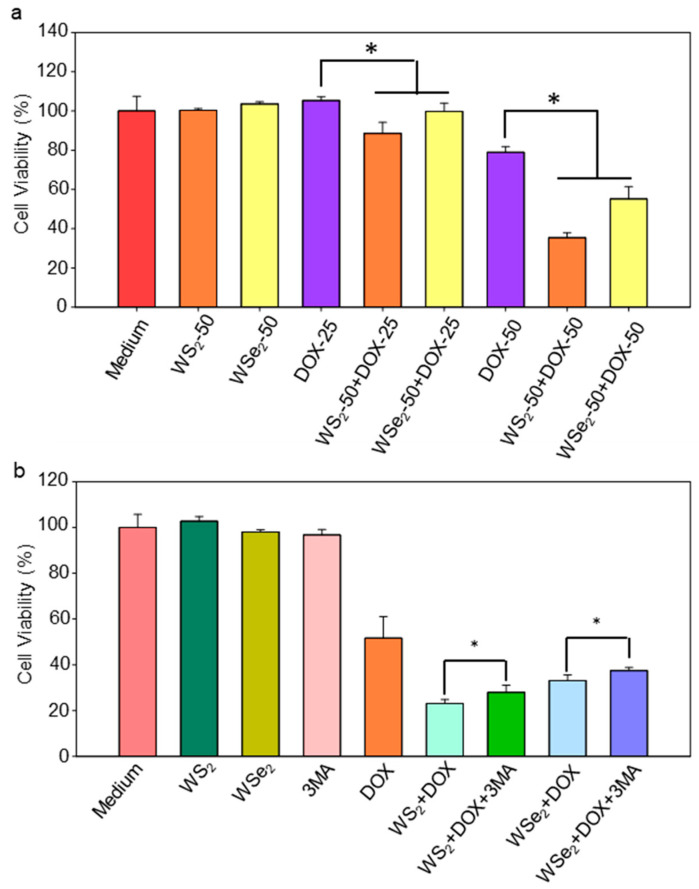
WS_2_ and WSe_2_ sensitized A549 cells to DOX treatment. (**a**) Pre-treatment with WS_2_ or WSe_2_ nanosheets enhanced the DOX-induced toxicity in A549 cells. Data are shown as the mean ± s.d.; n = 5; * *p* < 0.05. (**b**) Inhibition of autophagy partially reversed the enhanced sensitivity of A549 cells to DOX by WS_2_ or WSe_2_. A549 cells incubated with cell culture medium, WS_2_ (50 μg/mL), 3-MA (100 μM), or DOX (50 μM) for 24 h were the controls. Data are shown as mean ± s.d.; n = 5; * *p* < 0.05.

## Data Availability

Data is contained within the article and Appendix A.

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
