# Peer review of "Enhanced Sensitivity of A549 Cells to Doxorubicin with WS2 and WSe2 Nanosheets via the Induction of Autophagy"

_ijms, 2024, doi:10.3390/ijms25021164_

Round 1

Reviewer 1 Report

Comments and Suggestions for Authors

This is a high quality research in the rapidly expanding area of biologically active 2D-nanoparticles (nanosheets), which has emerged after the discovery of graphene twenty years ago. The work builds on previous experience of the Zhou group (references 25-27 in the manuscript), and the results are presented clearly and concisely. The most important message of the work, in my view, is that the nanosheets are more likely to adhere to the cell surface than penetrate the cells, and such adhesion can alter cell signaling by blocking receptors on the cell surface. The following minor changes are suggested:

(1)    The authors should comment on the potential for selective activity of the studied nanosheets in lung cancer cells, given that similar authophagy induction mechanisms were previously demonstrated in healthy bronchial cells (ref. 26). Will modification of nanosheets be required for selectivity?

(2)    There is a confusion between the main text and captions of Figures 6 and 7: the text states than the cells were pre-treated with nanosheets and the treatment medium removed before the addition of doxorubicin, while the figure captions mention co-exposure, which implies that the both drugs were added together. This point should be clarified.

(3)    Figure 6a does not add significant information to the verbal description in the text and can be removed. In addition, Figures 6 and 7 describe similar experiments, so they can be combined.

(4)    There is a formatting issue in the caption of Figure 2, which led to a large gap in the text.

(5)    In lines 2 and 3 (the title) and in line 149, the ‘2’ in WS2 and WSe2 should be in subscript.

(6)    Lines 74-75: change to ‘especially for malignant and multidrug-resistant tumors’.

(7)    Line 81: insert ‘of’ before ‘MoS2’.

(8)    Line 145: centrifugation speed and time should be specified.

(9)    Line 184: mention that the cytotoxicity data were presented in Supplementary Figure S1.

(10) Line 321: ‘expected’, not ‘respected’.

(11) Line 334: insert ‘nanosheets’ after ‘WSe2’.

Comments on the Quality of English Language

I've noticed a few grammatic errors (see above).

Author Response

Dear Editor,

Thank you very much for giving us an opportunity to revise our manuscript. We also appreciate the time and effort the reviewers have dedicated to reviewing our work. Now I am sending you a revised manuscript and a manuscript with tracked change for your review. Many insightful comments and suggestions from the reviewers helped us greatly improve our manuscript. We have made significant revisions according to comments and suggestions from all reviewers, which are marked in red in the revised manuscript. Here in below we address the reviewers’ comments point by point. We hope that it is now acceptable for publication. Thank you for your time and efforts.

We appreciate the reviewer for the positive comments and constructive suggestions on our manuscript. We have made significant revisions accordingly. The point-by-point responses were listed as followed.

Reviewer 2 Report

Comments and Suggestions for Authors

In the present study, the effect of WS2 and WSe2 nanosheets on doxorubicin (DOX) sensitivity was investigated using A549 lung cancer cells. The authors claimed that nanosheets can potentiate the cytotoxicity of DOX by autophagy induction.

Specific comments:

The study is too preliminary.

The authors used only one cancer cell line, namely A549 lung cancer cells. This is inadequate. More cancer cell lines must be used. Please note that it is a standard to use several cell lines and obtain the results to be published in a reputable scientific journal. Furthermore, the authors did not provide the information of the source of cell line. Cell line authentication test is required.

No characterization of WS2 and WSe2 nanosheets is provided. This is inadequate. The authors only cited the previous paper Zhou X, Jin W, Zhang R, Mao X, Jia J, Zhou H. Perturbation of autophagy pathways in murine alveolar macrophage by 2D TMDCs is chalcogen-dependent. J Environ Sci (China). 2024 Jan;135:97-107. doi: 10.1016/j.jes.2022.12.029. As the authors previously reported the nanosheet-induced autophagy, the novelty of the current study is also limited.

The involvement of autophagy in nanosheet-mediated augmentation of DOX-induced cytotoxicity is not convincing (Figure 7 and very slight effects of autophagy inhibitor 3MA).

As DOX can induce apoptosis in cancer cells, the authors should study the mode of cell death more comprehensively. The authors used only Cell Counting Kit-8 (CCK-8).

Moderate English language correction is needed.

Comments on the Quality of English Language

Moderate English language correction is needed.

Author Response

(The authors gave the same response as above.)

Round 2

Reviewer 2 Report

Comments and Suggestions for Authors

The authors did not substantially improve the paper according to my comments. The study is still preliminary. The authors stated “the preliminary nature of the research was intentional”. This is confusing. Additional data involving other cell lines are essential. More mechanistic studies are also needed. The better characterization is material is also needed. I do not recommend the paper in its current form.

Author Response

Reviewer: 2

Comments and Suggestions for Authors:

The authors did not substantially improve the paper according to my comments. The study is still preliminary. The authors stated “the preliminary nature of the research was intentional”. This is confusing. Additional data involving other cell lines are essential. More mechanistic studies are also needed. The better characterization is material is also needed. I do not recommend the paper in its current form.

We thank the reviewer for the advice. To demonstrate that the enhanced sensitivity of cancer cells to DOX mediated by WS2 and WSe2 can cross cell lines, human glioma cell line U87 cells were further validated.  

In line 345-350: “To demonstrate that the enhanced sensitivity of cancer cells to DOX mediated by WS2 and WSe2 can cross cell lines, human glioma cell line U87 cells were further validated. WS2 and WSe2 alone did not cause significant cell death at concentration of 50 μg/mL in U87 cells (Figure S3). Compared with U87 cells treated with DOX, cells treated with WS2 and WSe2 are more sensitive to DOX, and cell viability further decreased by 10-26% (Figure S4).”

We further validated the mode of cell death. Relative discussion was added accordingly.

In line 380-387: “To clarify the relationship between apoptosis and the enhanced sensitivity of A549 cells to DOX induced by WS2 and WSe2, we further investigated the activation of apoptosis signaling at molecular level. Apoptosis is a programmed cell death process controlled by a caspase signaling cascade. Caspase 3 is the central effector, and the cleavage of caspase 3 indicates the execution of apoptosis. We analyzed the cleavage of caspase 3 through Western blot. Our data showed that compared with cells treated with DOX, pre-treatment with WS2 and WSe2 induced more obvious cleavage of caspase 3 (Figure S5).”

Existing studies have shown that the surface charge of nanoparticles affects the cellular perturbation. Thus, we further measured the zeta potentials of WS2 and WSe2. The relevant characterization results and descriptions have been added accordingly.

In line 180-184: “Both WS2 and WSe2 nanosheets carried negative charges, as shown by the zeta potentials (Table S1). In water, the zeta potentials of WS2 and WSe2 are −39.0 ± 1.9 mV and −41.2 ± 1.5 mV. However, in cell culture medium containing 10% FBS, the zeta potentials of WS2 and WSe2 were shifted to −11.3 ± 1.2 and −11.7 ± 0.9 mV (Table S1). indicating the protein adsorption on nanosheet surface.”

Round 3

Reviewer 2 Report

Comments and Suggestions for Authors

The manuscript was partially improved and now can be accepted for publication.